# Organizational Determinants, Outcomes Related to Participation and Adherence to Cancer Public Health Screening: A Systematic Review

**DOI:** 10.3390/cancers17111775

**Published:** 2025-05-26

**Authors:** Daniela Amicizia, Maria Francesca Piazza, Federico Grammatico, Rosa Lavieri, Francesca Marchini, Matteo Astengo, Irene Schenone, Gabriella Paoli, Filippo Ansaldi

**Affiliations:** 1Regional Health Agency of Liguria (ALiSa), 16121 Genoa, Italy; daniela.amicizia@unige.it (D.A.); federico.grammatico@alisa.liguria.it (F.G.); rosa.lavieri@alisa.liguria.it (R.L.); francesca.marchini@alisa.liguria.it (F.M.); matteo.astengo@alisa.liguria.it (M.A.); irene.schenone@alisa.liguria.it (I.S.); gabriella.paoli@alisa.liguria.it (G.P.); filippo.ansaldi@alisa.liguria.it (F.A.); 2Department of Health Sciences (DiSSal), University of Genoa, 16132 Genoa, Italy; 3University Hospital of Cagliari, 09123 Cagliari, Italy

**Keywords:** cancer screening, organizational determinants, participation, adherence, public health programs, health equity, health system performance, digital health, community engagement, audit and feedback

## Abstract

Cancer screening programs for breast, cervical, and colorectal cancers save lives by detecting diseases early. However, many people do not participate regularly, limiting their effectiveness. This review analyzed studies published between 2015 and 2025 to understand how organizational factors influence screening participation. It found that successful programs share features like coordinated management, personalized invitations, and community involvement. Additionally, technology-based reminders, culturally adapted education, and quality monitoring improve screening attendance, especially among underserved groups. Effective screening programs require integrated approaches that combine clear communication, community trust-building, and digital tools. Policymakers should support structured, inclusive strategies to enhance participation and reduce health disparities, ultimately benefiting public health significantly.

## 1. Introduction

Cancer remains one of the leading causes of morbidity and mortality worldwide, posing a significant burden on individuals, health systems, and societies [1]. According to the World Health Organization (WHO), cancer caused nearly 10 million deaths in 2020, accounting for approximately one in six deaths [2]

It is considered that 30% to 50% of cancer cases are preventable through the effective use of prevention strategies; therefore, significant global efforts have been directed toward promoting health campaigns aimed at minimizing delays and overcoming barriers to timely cancer diagnosis. In line with this, the World Health Organization (WHO) advocates for the establishment of structured screening programs guided by standardized international protocols [2]. However, despite the availability of programs and a growing body of research and interventions focused on promoting cancer screening, participation rates remain suboptimal. This highlights the need for a thorough assessment of current practices, priorities, and obstacles in public health service delivery and research related to cancer prevention [3].

Public health screening programs for cancer, particularly for breast, cervical, and colorectal cancers, represent a cornerstone of secondary prevention strategies aimed at detecting disease at an early, more treatable stage. Indeed, cancer screening programs are valuable preventive tools that allow for early interventions, which can enhance outcomes [4]. Although population-based screening initiatives for cervical, colorectal, and breast cancer have been widely implemented across several countries, participation rates among eligible individuals remain suboptimal [5,6]. Additionally, while the clinical effectiveness of these screening programs has been extensively studied, comparatively less focus has been placed on the organizational frameworks that support their successful implementation [7]. The planning, coordination, and implementation of these programs are crucial for ensuring their effectiveness, equity, and sustainability [8].

Organizational approaches must be specifically tailored to address specific steps and interfaces within the screening process, ensuring that programs are effectively realized and accessible to all populations. The organizational aspects of oncologic screening encompass a broad range of elements, including governance models, integration within primary care, information systems, human resource management, population outreach, quality assurance, and follow-up mechanisms [9]. These components influence not only the uptake and coverage of screening but also its cost-effectiveness and public trust. Variability in how screening programs are organized—across different countries, regions, and healthcare systems—reflects differences in policy priorities, resource availability, and health system maturity. However, such variability also presents challenges in establishing best practices and ensuring equitable access to early detection services. Furthermore, knowledge and perceptions about cancer and screening practices can influence the decision to participate in cancer screening [10].

In recent years, the need for resilient and adaptive screening systems has become more evident, especially during the COVID-19 pandemic, which disrupted routine cancer screening services in many countries. Understanding the organizational enablers and barriers to maintaining and scaling up oncologic screening programs is, thus, critical for informing policy and guiding future investments.

This systematic review aims to synthesize current evidence on the organizational dimensions of public health cancer screening programs. By examining studies across diverse healthcare contexts, this review seeks to identify common structural features, successful implementation strategies, and potential pitfalls. Ultimately, it contributes to a deeper understanding of how organizational frameworks impact the effectiveness and equity of cancer screening in the population.

## 2. Materials and Methods

### 2.1. Study Design and Registration

The search strategy was developed following the PICO framework (Population, Intervention, Comparison, Outcome) [11], as recommended by the Cochrane Handbook for Systematic Reviews of Interventions. This systematic review was conducted in accordance with the PRISMA (Preferred Reporting Items for Systematic Reviews and Meta-Analyses) guidelines [12], and the protocol was registered in the PROSPERO database [13] (registration number: CRD420251029265).

A systematic literature search was conducted in the PubMed and Scopus databases to identify relevant studies published from January 2015 to January 2025. The search strategy aimed to retrieve articles focusing on cancer screening (specifically for breast, cervical, and colorectal cancer), organizational determinants, and outcomes related to participation and adherence.

The search string combined terms related to types of cancer screening (such as “cancer screening”, “breast cancer screening”, “mammography”, “cervical cancer screening”, “Pap smear”, “Human Papillomavirus (HPV) test”, “colorectal cancer screening”, “colonoscopy”, and “Fecal occult blood test (FOBT)”), organizational strategies (“organizational determinants”, “awareness campaigns”, “personalized invitations”, “recall system”, “health education programs”, “program implementation”, “health interventions”), and indicators of adherence or participation (“screening participation”, “compliance”, “uptake”, “engagement”, “acceptance”).

The search strategy was developed using a combination of text words and MeSH (Medical Subject Headings) terms depending on the database to capture some of the following concepts: cancer, screening, public health interventions, and screening adherence. The search strategy was peer-reviewed by a second information specialist in accordance with the Peer Review of Electronic Search Strategies checklist [14].

### 2.2. Study Selection Criteria

#### 2.2.1. Inclusion Criteria

To be included in the review, studies had to meet several predefined criteria. Eligible studies were those involving adult participants aged 18 years or older, particularly individuals falling within the recommended age ranges for breast, cervical, or colorectal cancer screening programs. Only studies evaluating organized or structured screening interventions were considered, with a specific focus on organizational strategies such as invitation systems, recall mechanisms, health education campaigns, or program implementation efforts aimed at improving participation or adherence.

In order to ensure the relevance of the findings, only studies that reported at least one quantitative measure related to participation, such as screening uptake, compliance with invitations, or test completion rates, were included. Eligible study designs encompassed both experimental and observational methodologies, including quasi-experimental studies, cohort and cross-sectional designs, and mixed-methods studies that provided quantitative participation outcomes. Studies had to be conducted in any healthcare or public health setting targeting the general screening-eligible population, and no restrictions were placed on geographical location. Furthermore, only peer-reviewed full-text articles written in English and published between January 2015 and January 2025 were considered for inclusion.

#### 2.2.2. Exclusion Criteria

Studies were excluded if they focused exclusively on pediatric populations, high-risk or vulnerable groups (e.g., individuals with a personal or family history of cancer), or participants outside the target age range for screening. Research that addressed opportunistic (non-organized) screening practices without incorporating structured organizational elements was also excluded. Additionally, studies were not eligible if they failed to report outcomes related to participation or adherence or if their primary focus was on knowledge, attitudes, beliefs, or perceptions without measuring actual screening behaviors.

Other exclusion criteria included publication types such as systematic, narrative, or scoping reviews, editorials, commentaries, protocols, conference abstracts, or letters to the editor. Randomized controlled trials were also excluded, as the review aimed to capture evidence from real-world program implementation and observational contexts. Finally, studies published in languages other than English, not available in full text, or addressing cancer types beyond the scope of this review (such as lung or prostate cancer) were excluded, as were studies published outside the defined time window.

#### 2.2.3. Information Sources

The search was conducted using the following databases: MEDLINE and Scopus. The search was updated on 31 January 2025.

#### 2.2.4. Data Extraction

Data from each included study were extracted using a predefined standardized form. Two reviewers independently performed the extraction of all key data items (e.g., study design, setting, population characteristics, details of the screening intervention or program, and outcomes related to participation) (Table 1). Any discrepancies or disagreements between the two data extractors were resolved through discussion and consensus.

The study selection process was independently conducted by two reviewers, each screening 100% of the records. The web-based tool Rayyan [15] was used to manage the screening process, including deduplication, title and abstract screening, and documentation of inclusion and exclusion decisions. In line with recommended practice, we recorded various study characteristics and results in summary tables to facilitate analysis (Table 1). No data imputation was performed for missing outcome values, and analyses were conducted using the available data only.

#### 2.2.5. Quality Assessment

The revised version of the Cochrane Risk of Bias tool (ROB 2) was employed to assess the Risk Of Bias In Non-randomised Studies of Interventions (ROBINS-I) for non-randomized studies [16]. The risk of bias assessment was independently completed for each study by two reviewers (DA and MFP). Discrepancies were resolved by discussion. The Robvistool was used to create a risk of bias plot [17] [Figure 2].

## 3. Results

### 3.1. Search Results and Characteristics of Included Studies

A total of 1270 records were initially identified in the search. After duplicates were removed, 1073 full-text articles were screened. After all screening and eligibility criteria were applied, 1047 studies were excluded, and 26 articles were ultimately included in the review (Figure 1). Figure 1 shows a flow diagram of the literature search strategy and the review process following PRISMA (Preferred Reporting Items for Systematic Reviews and Meta-Analyses) guidelines [12].

### 3.2. Overview of Organizational Determinants

The organizational strategies described in the included studies were heterogeneous and spanned multiple domains. To systematically analyze their influence on participation and adherence to cancer screening, results were categorized according to six key organizational factors: (1) governance models, (2) integration within primary care, (3) information systems, (4) human resource management, (5) population outreach, and (6) quality assurance and follow-up mechanisms [9]. The findings are presented by cancer type and interpreted through this organizational framework.

### 3.3. Breast Cancer Screening

Organizational strategies to improve breast cancer screening uptake included interventions across all six organizational domains.

Community-engaged approaches were particularly effective in underserved populations (population outreach). For example, a U.S. program (the Mi-MAMO patient navigation initiative) that partnered with community organizations achieved marked improvements in mammography compliance among underinsured women [18]. Similarly, a quasi-experimental study among African American women showed that tailored telephone counseling combined with navigation support (human resource management) significantly reduced missed appointments and increased screening completion with an adjusted odds ratio (OR) of approximately 3.9 [19].

In Hong Kong, a multi-component program employing ethnic minority community health workers and linguistically adapted materials led to substantial improvements in participation (population outreach, human resource management). The intervention, targeting multiple types of cancer, reported a 42% absolute increase in cervical screening and comparable gains in breast cancer screening [20]. These results emphasize the effectiveness of sustained, trust-based outreach models.

Technology-driven interventions have shown promise in improving mammography participation (information systems). A U.S. study implemented an email reminder system enhanced by reinforcement learning algorithms to personalize content. Over two years, 81.5% of women engaged with at least one email, 25% scheduled a mammogram, and 22% ultimately completed the screening [21]. Notably, this digital intervention achieved similar reach and engagement across diverse demographic subgroups, suggesting it can increase screening equitably without exacerbating disparities. These findings highlight that repeated, tailored messaging via electronic health records or patient portals can activate a significant portion of eligible women.

In contrast, simpler reminder strategies were less effective. A Veterans Health Administration program using electronic reminders and phone calls led to a negligible increase in mammography rates (+0.01 percentage points), indicating that single-channel approaches may be insufficient in established healthcare systems [22].

Programs addressing system-wide quality improvement also emerged. In Canada, participation in a provincial audit-and-feedback system (wherein primary care physicians received regular reports on their patients’ preventive screening status) was associated with a modest but significant increase in participation (integration within primary care, quality assurance). Patients whose physicians were enrolled in the program had higher odds of being up-to-date with mammography (OR~1.1) [23].

Region-wide audit programs have similarly shown potential. In Lombardy, Italy, a comprehensive audit based on the PRECEDE–PROCEED planning model led to improvements such as digitizing appointment systems, re-training staff, and engaging stakeholders. While immediate gains in screening uptake were limited, the audits identified 232 critical issues, more than half of which required systemic solutions (governance models) [24,25].

### 3.4. Cervical Cancer Screening

For cervical cancer screening, many interventions centered on reaching women who traditionally have low participation rates or face barriers to Pap testing, as well as leveraging technology to improve adherence.

Community outreach and education proved effective in specific underserved groups. In Norway, in-person educational sessions in immigrant women’s native languages led to a modest but significant increase in screening coverage among Pakistani and Somali women (population outreach, human resource management), from 46% to 51% in intervention areas compared to 44% to 45.5% in control areas [26].

In Hong Kong, a multi-year program combining multimedia talks, bilingual materials, and navigation assistance by trained community health workers led to a 42% absolute increase in screening among South Asian women (population outreach, human resource management, information systems) [20]. These successes reflect the importance of culturally tailored, sustained interventions that build community trust.

In India, challenges such as poor coordination between program levels, inadequate data management, and funding delays hindered scale-up efforts, pointing to the need for context-sensitive designs and stronger governance and monitoring (governance models, quality assurance) [27].

Similarly, Burus et al. [28] proposed a structured cancer needs assessment framework developed through a community-engaged, mixed-methods process in Kentucky. Their model identified multi-level determinants of cancer-related outcomes, emphasizing the importance of integrating community input and local data into program planning (governance models). This participatory approach ensured better alignment with population needs.

One innovative project in rural India addressed some of these issues by deploying a mHealth-supported screening prototype: trained community health workers and nurses used a mobile app to educate and screen women in remote villages [29]. Over 8600 individuals were screened through door-to-door campaigns; however, only 37% of screen-positive women returned for follow-up, highlighting that socio-economic obstacles (like travel, cost, and fear) still need to be overcome even after initial access is improved (human resource management, follow-up mechanisms). These studies underscore that in low-resource settings, strengthening organizational infrastructure (training, data systems, funding) and community trust are both critical to boosting cervical screening participation.

Technology-mediated interventions for cervical screening have shown mixed results, with effectiveness varying by the intensity and design of the approach.

A U.S. study found that a one-time SMS reminder had no impact on Pap test uptake (information systems) [30]. Women who received a reminder text were no more likely to obtain a Pap test than those who did not, suggesting that a standalone SMS was insufficient to change behavior in that context (especially without an easy way to act on the reminder).

By contrast, more robust e-health interventions demonstrated clear benefits. In an integrated health system in California, adding a behaviorally framed “nudge” reminder for cervical screening through the patient’s online portal led to higher appointment scheduling and completion rates for Pap smears [31]. In that retrospective study, the proportion of women who scheduled a Pap test via the online system was 2.9% with the nudge (versus 1.6% in a control group with standard reminders, *p* < 0.001), and the overall completion of Pap tests over the follow-up period was also higher in the nudge group (23.5% vs. 17.0%, *p* < 0.001). Although the absolute percentages might seem small, these gains were achieved across a large population and were sustained over two years, indicating a meaningful system-wide impact.

In Norway, a smartphone app (FightHPV) more than doubled screening attendance (HR 2.3), especially among long-term under-screeners (information systems) [32].

These findings highlight that digital strategies can not only increase screening rates but potentially improve early detection by reaching those most in need of screening.

On the other hand, ease of use and direct access are important considerations for any tech-based strategy. In an English pilot study, women aged 25–64 who were overdue for screening received weekly text invitations for six weeks with a link to download a smartphone app for easy appointment booking (information systems) [33]. About 11% of the women who received these messages went on to book a screening appointment within five months, a modest uptake. Notably, among those who did book an appointment, 72% did so using traditional phone or in-person scheduling methods, and only 28% booked through the app itself. This suggests that the text message prompt served as the primary catalyst to action, while the app convenience was utilized by a smaller subset. This may reflect a broader behavioral pattern: many individuals, especially those less digitally engaged, are more comfortable scheduling health appointments through direct contact rather than via apps, particularly when committing to a fixed date and time. This highlights the importance of designing digital interventions that accommodate varying preferences for interaction and reduce friction in the scheduling process.

The overall outcome still indicates that text-based outreach (even without universal app adoption) can nudge a portion of never- or overdue-screened women to complete their Pap test, though the reach was limited. Together, these studies indicate that technology can improve cervical screening adherence, but effectiveness hinges on multi-faceted engagement (combining reminders with accessible action steps or educational components). Simply informing patients via SMS may have little effect unless accompanied by user-friendly pathways to get screened (such as direct scheduling links, mailed self-sampling kits, or engaging educational content).

At the organizational level, results were varied. A multi-level initiative within the U.S. Veterans Affairs system slightly increased screening (by 0.9 percentage points) from an already high baseline [22]. This modest change illustrates how, in settings with established screening programs, additional organizational efforts may face diminishing returns unless they introduce novel elements or considerable resources.

In Lombardy (Italy), a region-wide audit improved organizational indicators but did not boost participation in the short term, suggesting that systemic change takes time to influence behavior (governance models, quality assurance) [24].

More encouraging results have been seen when organizational feedback is directly tied to providers. In Canada, audit-feedback tools used by family physicians led to a small but positive effect on cervical screening rate points (OR~1.06) [23] (integration within primary care, quality assurance). Even low-intensity efforts like performance feedback can incrementally improve preventive care delivery.

Finally, some studies focused on tools and frameworks. An Italian team developed a model for assessing the quality of health information on public websites related to cancer screening (information systems) [34]. By evaluating 46 local health organization websites, they formulated the “OSEC-p” framework to ensure online content about breast and cervical screening is clear, accurate, and engagement-oriented. While this study did not measure screening uptake directly, its intention was to improve how programs communicate with the public, which could, in turn, influence participation by improving knowledge and trust.

Likewise, an exploratory U.S. study synthesized qualitative and quantitative data from multiple cancer screening programs to create a conceptual model for integrating evidence-based interventions at different levels (patient reminders, provider prompts, electronic medical record optimization, etc.) across the phases of screening delivery (information systems) [35].

### 3.5. Colorectal Cancer Screening

A wide range of organizational interventions has been implemented to improve colorectal cancer (CRC) screening, particularly through fecal occult blood test (FOBT) and fecal immunochemical test (FIT) programs. Several studies compared direct mail of stool test kits to more personalized strategies. In a pilot study at a U.S. community clinic serving Latino adults, investigators tested two approaches side by side: an in-reach intervention (one-on-one interaction during a clinic visit, including a 30-min session with a patient navigator, an educational flip-chart, and handing the patient a FIT kit) versus an outreach intervention (mailing a FIT kit with culturally tailored instructions and a prepaid return envelope) (human resource management, population outreach) [36]. FIT completion within three months was 76% in the in-reach group compared to 19% in the mailed outreach group, despite similar follow-up protocols. This highlights the significant impact of personal contact and immediate education, although mailed outreach can still increase screening coverage on a larger scale when supported by other measures.

A large retrospective study in a Texas Federally Qualified Health Center network implemented a mailed FIT outreach program in English and Spanish, targeting over 33,000 average-risk adults [37]. The packet included an introductory letter, FIT kit, instructions, and a postage-paid return mailer, followed by text and letter reminders for non-responders and navigation support for colonoscopy if needed. The program achieved a 19.9% FIT completion rate, with 5.6% of returned tests being positive. Among those, 72.5% completed a diagnostic colonoscopy with the help of patient navigators, indicating strong follow-through. An equity-related finding was that Hispanic/Latinx patients, Spanish speakers, and uninsured individuals had higher FIT completion rates than other subgroups in this program. This contrasts with many screening initiatives where disadvantaged groups have lower uptake; in this case, the culturally and linguistically tailored materials and the safety-net setting likely facilitated better reach into those communities.

Together, Castaneda’s and Scott’s studies suggest that while mail outreach alone yields a moderate response, combining mailed screening kits with accessible education and patient navigation can substantially improve participation and ensure high completion of the screening cascade, particularly in minority and low-income populations.

These findings are supported by a large retrospective study conducted in Spain by Vanaclocha-Espi et al. [38], which examined over 1.7 million screening invitations across multiple regions. The study found that participation was significantly higher, reaching up to 68.6%, when recipients received FOBT kits automatically without needing to actively request them. In addition, quantitative immunological tests (FIT) led to greater participation than guaiac or qualitative tests, and women were more likely to participate than men across all age groups. These results reinforce the role of logistical and organizational choices, such as test delivery mode and type, in shaping screening uptake (governance models, information systems).

Beyond mail-based programs, multi-component interventions and novel reminder systems have been employed to enhance CRC screening, especially in groups with historically lower participation. A controlled interrupted time-series study in an urban safety-net health network assessed the effect of a multi-component intervention aimed at increasing FIT-based screening among adults aged 45–49 (integration within primary care, information systems) [39]. This younger age group had only recently become eligible for average-risk CRC screening under updated guidelines, and baseline screening rates were low. The intervention rolled out across 11 primary care clinics and included proactive mailing of FIT kits, coupled with text message and email reminders, and a standing order policy allowing medical assistants to give out FIT kits without a physician visit. Over the intervention period (~18 months), CRC screening completion in the 45–49 age group rose significantly faster than before; the completion rate increased by 2.8% per month post-intervention, compared to a 0.4% per month increase prior to the intervention. After adjusting for trends in a comparison group (aged 51–55, who did not receive the intervention), the net difference in monthly screening uptake slope was +1.7% (meaning the intervention added 1.7 percentage points to the monthly growth rate of screening completion) [39]. This translated into a substantial absolute improvement over time in the proportion of 45–49 year-olds screened, demonstrating the effectiveness of a coordinated, multi-modal strategy to engage a previously neglected age group.

In Australia, a study using the Health Action Process Approach (HAPA) model examined psychosocial factors influencing participation in stool-based CRC screening [40]. An online survey found that constructs like risk perception, outcome expectancies, planning, and self-efficacy explained about 50% of the variance in prior participation. Participants rated potential strategies, including personalized messages from doctors, small incentives, and helplines. Interventions that addressed both motivational and planning factors received the highest approval (population outreach, information systems) [40]. Although based on self-reported preferences, the study provides useful insights for designing comprehensive interventions.

On a broader scale, intrinsic organizational characteristics and continuous quality improvement efforts play a pivotal role in CRC screening outcomes. Financial and structural factors also mattered; centers that received a greater share of their revenue from capitated managed-care arrangements (often indicating a stronger emphasis on preventive care) had significantly higher screening uptake. Conversely, centers serving populations with higher social vulnerability had lower rates. For example, the percentage of homeless patients in a health center’s clientele was negatively associated with CRC screening performance (governance models, human resource management) [41]. These findings point to the importance of capacity and context: clinics with more providers per patient and stable funding streams can dedicate resources to preventive outreach, whereas those overwhelmed by social needs may struggle to achieve high screening coverage.

Quality improvement interventions have attempted to bridge these gaps. In Canada, the PCSAR audit-and-feedback program included CRC screening; physicians who enrolled and received periodic performance reports saw slight increases in patients completing fecal tests or colonoscopies (quality assurance, follow-up mechanisms) [23].

Similarly, in Lombardy, regional audits [24] that introduced digitalization and targeted outreach in low-performing areas expanded program coverage, especially among the eligible 50–74 population (quality assurance, follow-up mechanisms). Although the report noted only marginal changes in actual fecal test uptake immediately following the audits, the expansion in coverage and improvements in organizational processes are expected to facilitate higher participation in subsequent screening rounds.

Several studies provided insights into the implementation process of organizational interventions for CRC screening. A qualitative evaluation of a large U.S. health plan’s first year of a mailed FIT outreach program highlighted five key barriers (governance models, quality assurance) [42]: (1) program design issues, such as ensuring the mailed kits and instructions were user-friendly; (2) vendor coordination and logistics for kit mailing and lab processing; (3) patient engagement and communication hurdles, including addressing low health literacy; (4) reactions and satisfaction of clinic staff and leadership (whose buy-in is necessary for sustained program support); and (5) processes for tracking kit returns and results. By recognizing these implementation challenges early, the organization could refine its approach (for example, by improving communication materials and workflow integration) to hopefully increase the return rate of FIT kits in subsequent years.

Finally, an academic–health system partnership was established to co-develop a decision support tool for CRC screening (governance models, quality assurance) [43]. In this case, a university research team worked closely with a healthcare organization to design and implement a patient-facing tool integrated into practice aimed at patients who had received a physician recommendation for CRC screening. While the publication was descriptive, it outlined key inputs and processes for the partnership, such as stakeholder alignment, iterative design with end-user feedback, and attention to organizational fit, and documented the outputs (a functional decision aid embedded in clinic workflow).

A summary of included studies and interventions is reported in Table 1.

**Table 1 cancers-17-01775-t001:** Characteristics of included studies and organizational strategies (n = 26).

Author, Year	Country	Study Design	Sample	Setting	Objective	Measuring Tool	Type of Cancer	Organizational Strategies	Results
Zumba et al., 2024 [18]	USA	Retrospective observational study	944	Healthcare setting	To analyze how program mechanisms foster trust, engagement, and policy change	Community-engaged patient navigation program	Breast cancer	Community, organizational, and policy-level outcomes of the Mi-MAMO program over 7 years.	Increased compliance before the Mi-MAMO program.
Burus et al., 2024 [28]	USA	Case study	Fifty-one (residents of Kentucky who did not work in a healthcare profession)	Hospital setting	Outline of CNA framework and its application in Kentucky through a community-engaged, mixed-methods approach	Conceptual framework of multi-level determinants affecting cancer-related outcomes, focus group	Breast cancer screening and colorectal cancer	Online focus groups 2015–2021.	The 59-page report was broken down into five sections, including an executive summary of findings.
Relyea et al., 2023 [22]	USA	Descriptive observational study	8520	Veterans VA facilities	Conduct mixed-methods evaluation using the RE-AIM framework to assess the program implementation	RE-AIM (reach, effectiveness, adoption, implementation, and maintenance) framework	Cervical and breast cancer screening	Collaborative teams, tailored roles, improved communication, institutional support, and specialized training to enhance care for women veterans. Program implemented over three years (FY21–FY22).	The program grew by 50% and 117%, respectively. The program demonstrated effectiveness as screening rates increased for cervical and breast cancer screening +0.9% and +0.01%, respectively.
Conte et al., 2024 [34]	Italy	Pilot study	46 Italian websites	Italian local health organizations	Propose a framework (model) for the assessment of communication (through websites) aimed at breast and cervical cancer screening adherence	OSEC-p model	Breast and cervical cancer screening	Evaluation of health communication on websites, focusing on strategic orientation, stakeholder engagement, website ergonomics, and content related to cancer screening adherence. Data collected from May to June 2022.	Websites of Italian local health organizations scored an average of 58.18 out of 100, indicating moderate communication adequacy. The best-performing website achieved a score of 73.44, while the lowest scored 40.63. Findings highlighted weaknesses in content and ergonomics, suggesting areas for improvement to enhance communication effectiveness for cancer screening adherence.
Dsouza et al., 2022 [27]	India	Field observations combined with a key informant approach	Participants included 3 state program managers/coordinators, 11 district program managers/coordinators, 7 district hospital gynecologists/superintendents, 1 taluk gynecologist, 1 district oncologist, 7 NPCDCS staff, 7 CHC/PHC medical officers, 1 staff nurse, and 5 ANMs/ASHAs	Three States of India (Himachal Pradesh, Meghalaya, and Karnataka; seven districts each)	Consider the impact of different approaches to program organization, service delivery, and promotion of cervical cancer screening	Semi-structured interviews	Cervical cancer	Opportunistic screening at district hospitals, pilot projects testing different implementation strategies in Himachal Pradesh, Meghalaya, and Karnataka, screening integrated into the NPCDCS program initiated by the Government of India in 2010.	Participants perceive the existing capacities across the six domains as insufficient to implement the CCS program nationwide. Context-specific implementation, better coordination between the program and district health facilities, timely remuneration, better maintenance of data, and a strong monitoring system are possible solutions to remove health system-related barriers.
Subramanian et al., 2022 [35]	USA	Exploratory assessment using qualitative and quantitative data	Health systems and their partners, including federally qualified health centers (21 programs)	Health systems funded by the CDC’s Colorectal Cancer Control Program (CRCCP) and National Breast and Cervical Cancer Early Detection Program (NBCCEDP)	Describe how programs and their partners integrate evidence-based interventions (e.g., patient reminders) and supporting activities (e.g., practice facilitation to optimize electronic medical records) across colorectal, breast, and cervical cancer screenings	Conceptual model of three major categories: (1) multi-level interventions and supporting activities; (2) screening delivery phases; and (3) evaluation components. Site visits and follow-up telephone interviews	Colorectal, breast, and cervical cancer screenings	Integration of evidence-based interventions (e.g., patient reminders, provider reminders) and supporting activities (e.g., electronic medical record optimization) across multiple levels (individual, provider, health system, program, and community) from 2018 to 2019.	Integration of interventions can improve efficiency but poses challenges due to differing eligibility criteria, screening intervals, and service locations. Key determinants of success include intervention complexity, cost, implementation climate, and staff engagement. Systematic studies are needed to evaluate the effectiveness, cost-effectiveness, and sustainability of integrated approaches. Eight research priorities were proposed to address knowledge gaps.
Chuang et al., 2019 [41]	USA	Cross-sectional study; descriptive study from administrative data Uniform Data System (UDS) and from the Area Health Resource File (AHRF)	Fully operational U.S. health centers (956)	Health centers participating in HRSA’s Health Center Program	Examine organizational factors associated with cervical and colorectal cancer screening rates among health centers funded by the Health Resources and Services Administration (HRSA)	Predictors of cancer screening rates were organizational finances, staffing and infrastructure, patient population attributes, and location	Cervical cancer, colorectal cancer	Analysis of organizational finances, staffing and infrastructure, patient population attributes, and local context in 2015.	Organizational characteristics positively associated with cancer screening rates include provider/patient staffing ratios, electronic health record status, percentage revenue from public capitated managed care, and local primary care provider availability. The percentage of homeless patients was negatively associated with screening.
Baldwin et al., 2020 [42]	USA	Qualitative descriptive study	Ten in-depth interviews with staff and leaders from two health plans	Clinical setting	Provide critical information to help health plans understand how to best launch mailed FIT programs	Consolidated framework for implementation research; qualitative software program: Atlas.ti	Colorectal cancer	Collaborative Model: Health plan partnered with health centers to customize materials and workflows while coordinating FIT kit mailings. Centralized Model: Health plan executed all program elements internally. Implementation period: First year of the BeneFIT study. One year, 2017.	Challenges in five thematic areas: (1) program design, (2) vendor experience, (3) engagement/communication, (4) reaction/satisfaction of stakeholders, and (5) processing/returning of mailed kits.
Tabriz et al., 2020 [43]	USA	Descriptive study	Sixteen healthcare organization leaders and staff	Health care organization	Describe how a healthcare organization/university-based research partnership was developed and used to design, develop, and implement a practice-integrated decision support tool for patients with a physician recommendation for colorectal cancer screening	Case study approach, project documentation records, and semi-structured questionnaires	Colorectal cancer	Development and implementation of the e-assist: Colon Health program embedded in the electronic health record (EHR) and patient portal; multi-year partnership with shared decision-making, communication, and problem-solving processes.	Organization of the key inputs, processes, and outcomes of a healthcare organization/university-based research partnership.
Leigh et al., 2017 [23]	Canada	Retrospective cohort design	Included 7800 physicians; 1,206,660 men and women for colorectal screening. For breast and cervical screening, the population of women was 852,078 and 1,348,005, respectively	Primary Care Screening	Evaluate audit and feedback tools to determine effectiveness and to identify opportunities for improvement	Administrative databases	Colorectal, breast, and cervical cancer	PCSAR 2014; Two exposures were evaluated for each cohort: enrollment with a physician who was registered to receive the PCSAR and enrollment with a registered physician who also logged into the PCSAR.	Across all three screening programs, 63% of eligible physicians registered to receive the PCSAR, and 38% of those registered logged in to view it. Patients of physicians who registered were significantly more likely to participate in screening, with odds ratios ranging from 1.06 [1.04;1.09] to 1.15 [1.12;1.19]. PCSAR was associated with a small increase in screening participation.
Vanaclocha-Espi et al., 2017 [38]	Spain	Retrospective cohort study	Included 1,995,719 invitations—men and women aged 50 to 69 years	Community: Catalonia, the Valencian Community, Murcia, Cantabria, the Canary Islands, and the Basque Country	Identify and quantify the influence of certain organizational and sociodemographic factors, such as age, sex, municipality of residence, FOBT delivery type, type of FOBT, and screening history on participation rates	Invitation	Colorectal cancer	Organizational factors such as the type of FOBT delivery and the type of FOBT (guaiac and qualitative or quantitative immunological).	Included 1,748,753 invitations. Initial screening–first invitation group, participation was higher in women than in men in all age groups (OR 1.05 in persons aged 50–59 years and OR 1.12 in those aged 60–69 years). Participation was also higher when no action was required to receive the FOBT kit, independent of the type of screening (initial screening–first invitation [OR 2.24], subsequent invitation for previous never-responders [OR 2.14], subsequent invitation—regular [OR 2.03], subsequent invitation—irregular intervals [OR 9.38]) and when quantitative rather than qualitative immunological FOBT (FIT) was offered (initial screening–first invitation [OR 0.70], subsequent invitation for previous never-responders [OR 0.12], subsequent invitation—regular [OR 0.20]) or guaiac testing (initial screening–first invitation [OR 0.81], subsequent invitation for previous never-responders [OR 0.88], subsequent invitation—regular [OR 0.73]).
Bhardwaj et al., 2023 [30]	USA	Retrospective case-control study	Included 16,002 unique patient phone numbers	Institution’s registry of patients	Evaluating the effectiveness of a text messaging intervention on the uptake of cervical cancer screening at a single institution	Message	Cervical cancer	Single text message reminder 1 year.	Our text messaging intervention to improve Pap smear rates did not show a statistically significant difference between the intervention group receiving a text message and the control.
So et al., 2022 [20]	Hong Kong	Theory-based and culturally aligned trained program using a pretest/post-test study design	Not specified	Hong Kong Special Administrative Region (HKSAR)	To share strategies for improving ethnic minorities’ access to cancer screening services in Hong Kong and to illustrate the development and scaling up of the IMPACT project	Not specified	Breast, cervical, and colorectal cancers	Evidence-based multimedia interventions: Health talks, PowerPoint presentations, video clips, and distribution of health information booklets. Community health worker-led interventions: Training South Asian women to become community health workers, providing multimedia education, follow-up calls, and navigational assistance.	Significant increase in cancer screening uptake among South Asians (e.g., 42% increase in cervical cancer screening uptake over 5 years). Feasibility and effectiveness of multimedia and community health worker-led interventions. Positive feedback from community health worker trainees and suggestions for improvement. Policy impact: Legislative councilor cited findings to advocate for health policies for ethnic minorities.
McClellan, et al., 2024 [39]	USA	Controlled interrupted time series (ITS) analysis	Included 7816 unique patients (3873 aged 45–49 and 3943 aged 51–55)	Eleven primary care clinics in the San Francisco Health Network, an urban safety-net health system	Assess the effect of a multi-component intervention on colorectal cancer (CRC) screening completion in patients aged 45–49	Electronic health record (EHR) data	Colorectal cancer	Multi-component intervention including mailed fecal immunochemical test (FIT), text messaging, email outreach, and standing order protocol for FIT; duration from 10 October 2021 to 2 May 2023.	The intervention increased CRC screening completion among patients aged 45–49, with an average increase of 2.8% every 30 days post-intervention rollout compared to 0.4% pre-intervention. The difference persisted after accounting for changes in the comparison group (slope difference 1.7%).
Madleen Orumaa et al., 2022 [32]	Norway	Retrospective Cohort Study	Included 4518 women aged 20–69 years (658 intervention group, 3860 reference group)	Norwegian Cervical Cancer Screening Program	To examine the impact of exposure to the FightHPV mobile app on cervical cancer screening attendance	Cumulative incidence and hazard ratios (HRs) with 95% CIs	Cervical Cancer	Exposure to the FightHPV app; follow-up period of 1 year.	Women exposed to the FightHPV app were 2 times more likely to attend screenings (adjusted HR 2.3, 95% CI 2.0–2.7) and 13 times more likely to be diagnosed with high-grade abnormality (adjusted HR 12.7, 95% CI 5.0–32.5) compared to the reference group.
Myers et al., 2022 [40]	Australia	Cross-sectional survey	Included 377 participants aged 50–74 years	Online survey via Qualtrics	To develop and test interventions to increase participation in mail-out bowel cancer screening using the Health Action Process Approach (HAPA)	Process Approach to Mail-out Screening (PAMS) scale and User Ratings of Mail-Out Screening Interventions (UR-MSI) scale	Bowel cancer	The study investigated various intervention strategies, including delivering messages with the FOBT kit, providing provisions with the screening invitation, and offering services alongside the NBCSP.	The HAPA model explained 49.9% of the variation in FOBT screening participation. Positive ratings of interventions ranged from 20.47% to 72.25%. Interventions targeting all HAPA factors are recommended to increase participation rates.
Scott et al., 2022 [37]	USA	Retrospective cohort study within a single-arm intervention	Included 33,606 patients aged 50–75	Central Texas Federally Qualified Health Center (FQHC) system	To examine the uptake and equity of a mailed stool test program for colorectal cancer screening	Electronic health records (EHR)	Colorectal cancer	Mailed outreach packets in English/Spanish, including introductory letter, free fecal immunochemical test (FIT), lab requisition, postage-paid mailer, instructions, and medical records update postcard; reminders via text and letter; bilingual patient navigator for follow-up colonoscopy.	Of 19.9% completed mailed FIT, 5.6% tested positive; 72.5% of positive FIT completed colonoscopy; higher completion rates among Hispanic/Latinx, Spanish-speaking, and uninsured patients.
Qureshi et al., 2021 [26]	Norway	Community-based intervention non-randomized trial	Included 10,820 women aged 25–69	Four geographical areas surrounding Oslo	Increase participation in cervical cancer screening among Pakistani and Somali women	Screening status obtained from the Norwegian Cancer Registry	Cervical cancer	Oral presentation in Urdu and Somali, practical information on appointment and payment, 20–25 min, conducted from February to October 2017.	Intervention group showed a significant increase in screening participation (from 46% to 51%) compared to the control group (from 44% to 45.5%). Absolute difference in change in proportion screened was 0.03 (95% CI; 0.02–0.06).
Bucher et al., 2022 [21]	USA	Retrospective, single-arm, observational study	Included 139,164 women aged 49.5 to 74 years	Large Catholic health system in the midwestern United States	To establish the feasibility of a reinforcement learning-enabled mammography digital health intervention delivered via email and to understand the intervention’s reach and ability to elicit behavioral outcomes of scheduling and attending mammograms across different demographic subgroups	Behavioral science-based email messages assembled and delivered by a reinforcement learning model	Breast cancer	Eligible individuals received up to 40 emails during the 2-year study period, with messages sent once per week for 5 weeks, followed by an 8-week pause, and then another pulse of one message per week for 5 weeks.	A total of 81.52% of women engaged with at least one email, 24.99% scheduled mammograms, and 22.02% attended mammograms (88.08% attendance rate among women who scheduled appointments). The intervention showed proportionate reach and engagement across diverse demographic subpopulations, suggesting it may equitably drive mammography uptake.
Odelli et al., 2024 [24]	Italy	Systematic audit evaluation	Approximately 10 million residents in Lombardy region, age cohorts for screening: 45–74 for breast cancer, 50–74 for colorectal cancer, 25–64 for cervical cancer	Lombardy region, Italy	To evaluate the impact of PRECEDE–PROCEED model audits on cancer screening programs in the Lombardy region, focusing on equity and quality improvement	Structured analysis methodologies, including epidemiological, behavioral, and organizational assessments	Breast, colorectal, cervical	Systematic region-wide audit performed in 2019, follow-up audits in 2022–2023; digitization of processes, stakeholder engagement, continuous re-training, targeted equity interventions.	Increased coverage for breast and colorectal screenings, slight decline in participation rates and examination coverage, notable improvements in organizational aspects, gaps in training, and equity-targeted actions remained.
Cereda et al., 2020 [25]	Italy	Development and first application of an audit system	Not specified	Lombardy, Italy	To describe the process of defining and testing a planning software application and an audit cycle based on the PRECEDE-PROCEED model to improve breast cancer screening	Planning software application based on the PRECEDE-PROCEED model	Breast cancer	Implementation of a peer-to-peer audit system and a software application to help plan interventions to improve screening programs at the local level; audit cycle with site visits, report generation, and monitoring at 3, 6, 9, and 12 months.	The plans produced using the application were more standardized and had clearer indicators for monitoring and evaluation compared to those produced in the previous year. The first round of audits identified 232 critical issues, with 53% of solutions to be activated for organizational critical issues.
Highfield et al., 2015 [19]	USA	Quasi-experimental design using type 1 hybrid design	Included 198 African American women aged 35–64	Mobile mammography provider in Houston, TX	Evaluate the effectiveness of an adapted mammography evidence-based intervention (EBI) in improving appointment keeping for mammography in African American women and describe processes of implementation in a practice setting	Logistic regression and intent-to-treat analysis	Breast cancer	Tailored telephone counseling reminders based on the Transtheoretical Model of Change, including needs assessment, barrier scripts, active listening, and training of patient navigator.	The intervention group had a significantly lower no-show rate (19%) compared to the control group (44%). Adjusted odds of attending the appointment were 3.88 (*p* < 0.001) for the intervention group versus 2.31 (*p* < 0.05) in the intent-to-treat analysis. Positive patient feedback on the intervention calls was reported, with high satisfaction ratings.
Castaneda et al., 2018 [36]	USA	Pilot Study	Included 200 Latino adults aged 50–75 years	Federally-Qualified Health Center (FQHC) in San Diego, CA	To test the implementation of two evidence-based intervention strategies to promote colorectal cancer (CRC) screening among Latino adults in a primary care setting	Fecal immunochemical test (FIT) completion and return within three months assessed through electronic medical records	Colorectal cancer	In-reach intervention: Opportunistic clinic visit including a 30-min session with a patient navigator, review of an educational flip-chart, and a take-home FIT kit with instructions. Outreach intervention: Mailed materials including FIT kit, culturally and linguistically tailored instructions, and a prepaid return envelope.	Results: In-reach intervention: 76% screening completion rate. Outreach intervention: 19% screening completion rate. Follow-up: Both interventions included follow-up calls to promote screening completion and referrals for additional screening and treatment if needed.
Liang et al., 2022 [31]	USA	Retrospective observational study	Medicare wellness visits: 43,889 patients (mean age 75 years); Pap smear: 288,152 patients (mean age 41 years)	Sutter Health, Northern California	To examine the impacts of behavioral economics-based nudge health maintenance reminders on appointment scheduling through a patient portal and appointment completion for Medicare wellness visits and Pap smear	Electronic health record data	Cervical cancer	Behavioral economics-based nudge health maintenance reminders implemented for Medicare wellness visits in November 2017 and for Pap smears in February 2018; analyzed data from January 2017 to February 2020.	Intervention vs. control: Higher rates of appointments scheduled through the patient portal for nudge reminders (Medicare wellness visits: 13.0% vs. 9.7%; Pap smear: 2.9% vs. 1.6%; *p* < 0.001). Adherence: Higher appointment completion rates for Pap smear (nudge: 23.5% vs. control: 17.0%; *p* < 0.001); comparable rates for Medicare wellness visits (nudge: 51.5% vs. control: 51.8%; *p* = 0.30). Satisfaction: Not directly measured. Follow-up: Sustained effect over time for scheduling appointments through the patient portal. Policy impact: Sutter Health implemented behavioral economics-based language for all health maintenance reminders on 28 May 2020.
Bhatt et al., 2018 [29]	India	mHealth-supported screening intervention	Included 8686 people screened (ages not specified)	Rural and remote communities in India (RUHSA, Mungeli, Padhar)	To determine the key features of an ideal mHealth prototype for cancer screening in LMIC settings, assess feasibility and acceptability, and evaluate the response to screening invitations	mHealth prototype with SIM card application	Cervical and oral cancer	Screening delivered by community health workers (CHWs) and nurses; training workshops held at each site; pilot testing; continuous feedback loop for refinements; 23-month evaluation period.	A total of 8686 people were screened, 98% for oral cancer. Positivity rates: 28% for cervical cancer and 5% for oral cancer. Follow-up attendance: 37% for cervical cancer and 31% for oral cancer. mHealth prototype improved efficiency and increased staff motivation. Significant barriers to follow-up due to socio-economic factors. Positive social impact on CHWs’ standing in communities
Ryan et al., 2020 [33]	England	Service Evaluation	Included 632 eligible women aged 25–64	Three general practice surgeries in a deprived East London borough	To assess the feasibility of offering women who are overdue for cervical screening the use of a smartphone app to book their appointment	Text message reminders and app-based booking system	Cervical cancer	Text messages sent in weekly batches over six weeks, inviting women to download an app to book their screening appointment.	A total of 11% of women with valid phone numbers booked a screening appointment within five months; 72% booked using standard methods, and 28% booked via the app. The text message reminder was likely the key active ingredient for most women.

### 3.6. Quality Assessment

Risk of bias assessments for the included studies are shown in Figure 2. Briefly, 38% (10/26) of the included non-randomized studies were classified as high risk, 58% (15/26) as having some concerns, and the remainder (1/26, 4%) were classified as low risk.

Some studies were rated at high risk of bias due to the absence of a control group, the use of self-reported cross-sectional data, and the lack of adjustment for potential confounders. In quasi-experimental designs, large baseline differences between groups and non-random allocation limited the comparability of intervention arms. Additional concerns included the use of different recruitment methods across groups, unequal access to resources such as insurance, and the absence of statistical comparison, all of which may have introduced confounding, selection bias, and reduced internal validity.

## 4. Discussion

This systematic review examined how organizational determinants affect participation and adherence in population-based screening programs for breast, cervical, and colorectal cancers. While the clinical efficacy of these screenings is well documented [44,45], uptake remains suboptimal in many countries due to a combination of system-level, provider-level, and contextual barriers [46,47].

Our findings reinforce a growing international consensus that improving cancer screening requires more than clinical or technological innovation; it demands the optimization of health service delivery systems [48,49]. Programs that are organized, and characterized by defined eligibility criteria, active invitation, centralized coordination, quality assurance mechanisms, and linkages to follow-up are consistently associated with higher participation and better equity outcomes compared to opportunistic models [50,51,52].

For example, the experience in Lombardy, Italy, showed how the implementation of a structured organizational assessment framework helped identify and address local barriers to participation in cervical cancer screening [34]. Although the immediate effect on adherence was limited, this strategy enabled the gradual alignment of protocols, improved communication, and the strengthening of audit and feedback systems.

Interventions incorporating community engagement, personalized communication, and integrated digital tools showed the most consistent success across breast, cervical, and colorectal cancer programs. Community-based navigation and culturally adapted outreach significantly improved participation, particularly among underserved and minority populations, as demonstrated by Zumba et al., So et al., and Qureshi et al.[18,20,26]. Similarly, tailored health education and the inclusion of trusted community figures were essential for establishing trust and ensuring program relevance.

Additionally, studies from the United States, India, and Italy demonstrate that community engagement strategies must be accompanied by strong organizational infrastructure, including training, supervision, and adequate funding, to sustain their effectiveness and scalability [27,29,34]. Programs that relied solely on local actors without systemic support often struggled with follow-up care and data management, limiting their long-term impact.

Technology-enhanced strategies, such as the reinforcement learning email system evaluated by Bucher et al. [21] and the interactive FightHPV mobile app studied by Orumaa et al. [32], demonstrated substantial increases in participation. However, simpler interventions, like a single SMS reminder tested by Bhardwaj et al.[30], were often ineffective in isolation, confirming the necessity of multi-faceted designs that include actionable follow-up steps.

The success of digital interventions appears closely tied to their integration within broader organizational ecosystems. For example, the FightHPV app was more impactful when embedded in a national screening registry [32], and nudge messaging through electronic health portals yielded measurable gains only when patients could directly act on reminders via appointment systems [31]. These findings suggest that digital tools alone are insufficient and must be coupled with accessible, user-friendly pathways to care.

Since individuals receiving abnormal results may often be new to screening, it is crucial to combine timely, clear communication with proactive counseling to mitigate anxiety and ensure appropriate follow-up. Supplementing traditional methods with digital tools, such as text messages or app-based notifications, can improve reach and effectiveness, especially among those unfamiliar with the screening process.

Audit and feedback mechanisms at the provider level, as seen in the work of Leigh et al. [23] and Odelli et al. [24], led to modest improvements in adherence, particularly when they were embedded into broader quality improvement cycles. These findings suggest that while systemic enhancements may not yield immediate changes in behavior, they contribute to a more efficient and equitable screening infrastructure over time.

Furthermore, implementation fidelity and contextual fit emerged as critical moderators of success. As noted by Subramanian et al. (2022), multi-component interventions that aligned with provider workflows and institutional goals were more likely to be adopted and sustained [35]. Similarly, integration into existing health information systems enhanced data tracking, reduced fragmentation, and facilitated real-time performance monitoring [42,43].

### 4.1. Differences Across Cancer Types

Breast cancer screening initiatives appeared especially responsive to multi-component outreach and system-level quality improvement. For cervical cancer, digital and culturally sensitive strategies (e.g., So et al. [20], Liang et al. [31]) were more effective in reaching under-screened women. In colorectal cancer programs, direct mailing of fecal tests accompanied by clear instructions and follow-up support, as shown by Scott et al.[37] and Castaneda et al. [36], improved both test return rates and diagnostic follow-through.

However, variability in outcomes across countries also reflects broader systemic differences. In high-income settings, screening initiatives often benefited from centralized coordination and reimbursement mechanisms, while in lower-resource contexts, fragmentation and underfunding constrained implementation despite strong community engagement [27,29,38].

### 4.2. Equity Considerations

A number of studies underscored the dual potential of organizational interventions: they can either reduce or exacerbate disparities. Positive equity outcomes were observed in programs designed with linguistic and cultural inclusivity in mind, such as those by So et al. [20] and Vanaclocha-Espi et al. [38], which reported higher participation among ethnic minorities when appropriate materials and delivery modes were used. Conversely, interventions implemented in low-resource contexts, like those described by Dsouza et al. [27] and Bhatt et al. [29], highlighted persistent challenges related to infrastructure, funding, and follow-up capacity, even when community health workers or mHealth tools were involved.

These findings underscore the need for equity-focused design from the outset. Interventions that consider social determinants of health, such as transportation, digital access, and health literacy, are better positioned to close participation gaps. For instance, programs using linguistically tailored communication and auto-delivery of screening kits showed higher success among socioeconomically disadvantaged groups [36,37,38].

### 4.3. Organizational and Structural Determinants

This review also demonstrates that health system maturity and organizational readiness are pivotal. Characteristics such as stable funding streams, optimal provider/patient ratios, advanced data systems, and dedicated preventive care mandates were positively associated with higher screening participation. Conversely, high social vulnerability and poor integration between governance levels, especially in decentralized or underfunded regions, limited program reach and continuity.

### 4.4. Limitations

The review relied heavily on observational data, with nearly 40% of included studies classified as high risk of bias due to weak design, absence of control groups, or unadjusted analyses. Additionally, heterogeneity in program definitions, outcome reporting, and intervention descriptions limits the generalizability of specific findings.

### 4.5. Implications for Policy and Practice

Policymakers and public health authorities should prioritize the implementation of multi-level interventions that combine organizational quality assurance with patient-centered outreach. Embedding audit-feedback cycles, standardizing protocols, and investing in digital infrastructure can enhance both efficiency and equity. Furthermore, the integration of culturally competent communication and accessible technologies is vital to reaching diverse populations and avoiding unintended exclusions.

Future policy frameworks should also emphasize cross-sectoral collaboration, linking cancer screening with primary care, social services, and community organizations. Moreover, international partnerships can support the adaptation of successful models across diverse health systems, particularly in low- and middle-income countries where organizational capacity is still developing [7,35,48].

## 5. Conclusions

Our research highlights that well-organized cancer screening programs, characterized by active invitation, centralized coordination, quality assurance, and clear follow-up mechanisms, are more likely to achieve higher participation rates and better equity outcomes compared to opportunistic models.

A critical factor influencing the effectiveness of organized screening programs is their multi-faceted, community-based approach. Interventions that combine personalized communication, community engagement, and the strategic use of digital tools have been shown to significantly improve participation, particularly among underserved and minority populations.

As expected, digital tools, such as mobile applications and reinforcement learning emails, can enhance screening participation among eligible individuals. However, simple, standalone technological interventions tend to be ineffective unless integrated into broader, action-oriented strategies.

Although provider-level audit and feedback mechanisms may not always lead to immediate behavior change, when embedded within larger quality improvement initiatives, they contribute to long-term gains in the efficiency and equity of screening programs.

It is important to recognize that cancer screening is a public health intervention that must be prioritized. By making screening processes clearer and more accessible to all, these programs can help reduce health disparities and build trust within communities.

A critical component of reducing cancer disparities is ensuring that cancer is detected at its earliest stages when treatment is most effective. Thus, it is essential to address the health-related social needs of communities and innovate with intention, ensuring that everyone has equitable access to cancer prevention and early detection services.

Policymakers and public health leaders should focus on advancing multi-level strategies that integrate organizational quality assurance with patient-centered outreach efforts. Incorporating audit and feedback mechanisms, standardizing procedures, and strengthening digital infrastructure are key actions to improve both the efficiency and fairness of screening programs.

Equally important is the use of culturally sensitive communication and accessible technologies to effectively engage diverse populations and prevent inadvertent exclusion.

Looking ahead, policy frameworks should promote cross-sector collaboration by connecting cancer screening initiatives with primary care, social services, and community-based organizations. In addition, fostering international cooperation can help adapt proven models to a variety of health systems, particularly in low- and middle-income countries where organizational capabilities are still emerging.

## Figures and Tables

**Figure 1 cancers-17-01775-f001:**
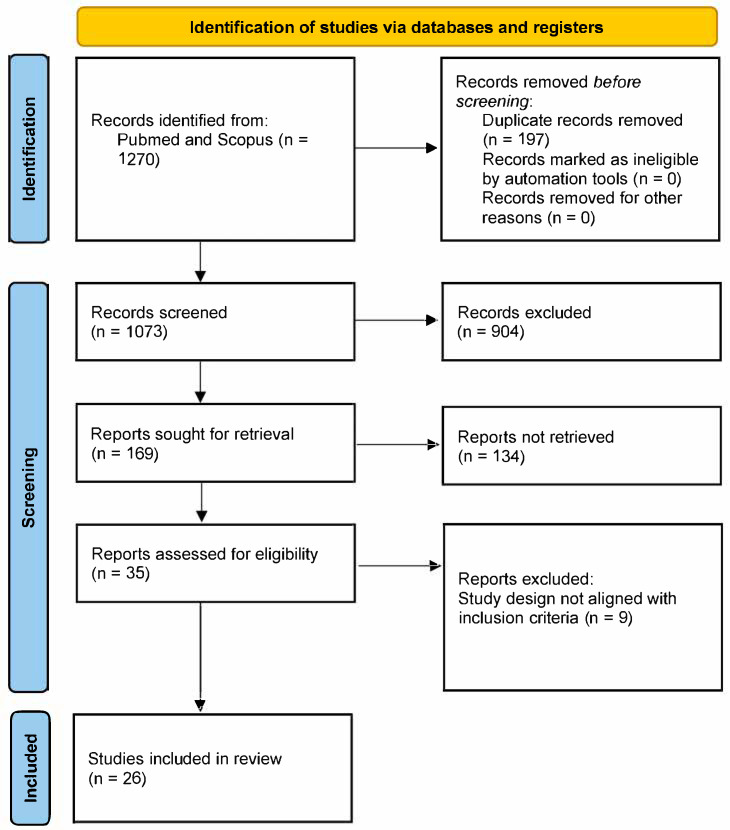
PRISMA (Preferred Reporting Items for Systematic Reviews and Meta-Analyses) diagram outlining the steps involved in identifying screened and included studies.

**Figure 2 cancers-17-01775-f002:**
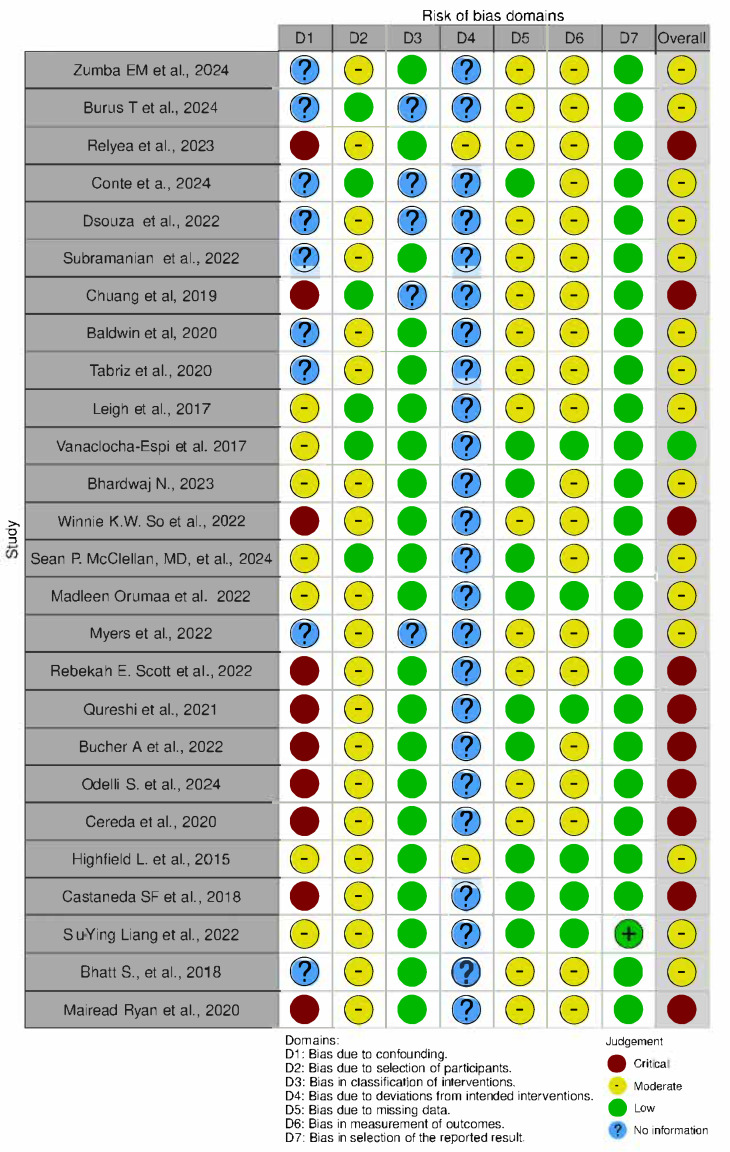
Risk Of Bias In Non-randomized Studies—of Interventions (n = 26) created using the Robvis tool [15].

## Data Availability

Data are included within the article.

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
