# Peer review of "Organizational Determinants, Outcomes Related to Participation and Adherence to Cancer Public Health Screening: A Systematic Review"

_cancers, 2025, doi:10.3390/cancers17111775_

Round 1

Reviewer 1 Report

Comments and Suggestions for Authors

The review titled “Organizational determinants, outcomes related to participation and adherence to cancer public health screening: a Systematic Review” aimed at understanding how organizational frame works impact the effectiveness and equity of cancer screening. This review is informative, however, the manuscript has too much redundancy and needs to be summarized and simplified. In addition, the layout of the tables needs to be changed. Here are some comments:

  1. Line 69-72, the author believes that the organizational aspects of cancer screening include 6 major factors. Please explain how these six factors affect cancer public health screening. Since this classification is very relevant to the topic, it is recommended to classify the screening of each type of cancer in the third part according to these six factors.
  2. Please add numbering to Section 3, e.g. 3.1, 3.2, etc. to make the article structure clearer.
  3. Each subsection in the 3. Results section is too long. The author basically uses a large paragraph to introduce each reference. This makes the entire review seem too redundant and the key information is unclear. Please simplify this section.
  4. The tables in the manuscript is not satisfactory. For example, half of page 17 is blank. Please adjust the spacing between each part of the table header to coordinate the layout of the table content. If necessary, delete the redundant description in the results section.
  5. Line 660, if this manuscript does not involve patents, please delete this subheading.

Author Response

Reviewer 1

Comments and Suggestions for Authors

The review titled “Organizational determinants, outcomes related to participation and adherence to cancer public health screening: a Systematic Review” aimed at understanding how organizational frame works impact the effectiveness and equity of cancer screening. This review is informative, however, the manuscript has too much redundancy and needs to be summarized and simplified. In addition, the layout of the tables needs to be changed. Here are some comments:

- Line 69-72, the author believes that the organizational aspects of cancer screening include 6 major factors. Please explain how these six factors affect cancer public health screening. Since this classification is very relevant to the topic, it is recommended to classify the screening of each type of cancer in the third part according to these six factors.

Thank you for your insightful comment. As suggested, we have revised the Results section (Section 3) to classify the findings according to the six organizational aspects outlined in lines 69–72: (1) governance models, (2) integration within primary care, (3) information systems, (4) human resource management, (5) population outreach, and (6) quality assurance and follow-up mechanisms. This classification has been systematically applied to the description of interventions for breast, cervical, and colorectal cancer screening. Each organizational factor is explicitly identified in relation to the relevant study findings, thus allowing for a clearer interpretation of how these domains influence screening participation and adherence. We believe this restructuring strengthens the conceptual framework of the review and aligns closely with the aims of the manuscript.

- Please add numbering to Section 3, e.g. 3.1, 3.2, etc. to make the article structure clearer.

Thank you for your suggestion. As requested, we have added sub-section numbering (e.g., 3.1, 3.2, etc.) to Section 3 to improve the clarity and navigability of the article structure. Please see the revised Section 3 for reference.

- Each subsection in the 3. Results section is too long. The author basically uses a large paragraph to introduce each reference. This makes the entire review seem too redundant and the key information is unclear. Please simplify this section.

Thank you for your valuable feedback. As requested, I have revised section 3 by simplifying the text, breaking down the content into shorter paragraphs, and reducing redundancy to improve clarity and focus.

- The tables in the manuscript is not satisfactory. For example, half of page 17 is blank. Please adjust the spacing between each part of the table header to coordinate the layout of the table content. If necessary, delete the redundant description in the results section.

We have thoroughly revised the formatting of all tables to address layout issues. Specifically, we eliminated excessive white space and improved header alignment to enhance readability. We also removed redundant narrative descriptions from the results section where the information is already present in the tables.

- Line 660, if this manuscript does not involve patents, please delete this subheading.

We confirm that this manuscript does not involve any patents. The “Patents” subheading has been removed from the final version and as requested by the other reviewer we have corrected this heading and replaced “Patents” with “Contributions

Reviewer 2 Report

Comments and Suggestions for Authors

Amicizia et al studied the determining factors and outcome of public health screening of breast, cervical and colorectal cancer patients. They followed the PRISMA principle to write a systematic review and selected 26 studies for review. They concluded that organizational strategies are important factors to enhance screening participation and reduce disparities.

The study is performed well and they presented their results nicely. However, there are some concerns that need to be edited and modified.

  1. In author list, * should be mentioned as corresponding author.
  2. Line 107, HPV and FOBT should be expanded.
  3. The sentence in line 282-284, author should emphasize the importance of counselling should be employed to convince them . Also, as they appeared for the first time, the abnormal should be heavily communicated by digital or other supplementary methods.
  4. Line 318 and 324-328, many people still do not rely APP for scheduling an appointments, which has a consequence (physical attendance in a particular date and time), rather than retrieving information.
  5. Line 370, FIT should be expanded
  6. In line 405, author should provide a percentage that is significantly higher.
  7. Line 660, it should be “Contributions” instead of “Patents”

Author Response

Amicizia et al studied the determining factors and outcome of public health screening of breast, cervical and colorectal cancer patients. They followed the PRISMA principle to write a systematic review and selected 26 studies for review. They concluded that organizational strategies are important factors to enhance screening participation and reduce disparities.

The study is performed well and they presented their results nicely. However, there are some concerns that need to be edited and modified.

- In author list, * should be mentioned as corresponding author.

Thank you for your suggestion. We have added an asterisk (*) to identify the corresponding author in the author list.

- Line 107, HPV and FOBT should be expanded.

This has been corrected. We expanded the abbreviations upon first mention as follows: HPV – Human Papillomavirus, FOBT – Fecal Occult Blood Test

- The sentence in line 282-284, author should emphasize the importance of counselling should be employed to convince them . Also, as they appeared for the first time, the abnormal should be heavily communicated by digital or other supplementary methods.

We thank the reviewer for the valuable comment. We agree that counseling plays a crucial role in improving screening adherence, especially among individuals who are undergoing screening for the first time or who receive abnormal results. In response to the suggestion, we have revised the sentence in lines 282–284 to more clearly emphasize the need for personalized counseling strategies in discussion section. We also highlight the importance of integrating digital and supplementary communication methods to ensure that individuals with abnormal results are adequately informed and supported.

- Line 318 and 324-328, many people still do not rely APP for scheduling an appointments, which has a consequence (physical attendance in a particular date and time), rather than retrieving information.

We thank the reviewer for the insightful comment. We agree that many individuals, particularly in certain demographic groups, still prefer traditional methods for scheduling appointments due to familiarity, trust, or ease of access. This limitation can indeed affect the effectiveness of app-based systems, especially when scheduling requires committing to a specific date and time rather than simply retrieving information. In response, we have revised the corresponding paragraph to explicitly acknowledge this issue and clarify its implications for digital intervention design.

- Line 370, FIT should be expanded

Thank you. FIT is now expanded as Fecal Immunochemical Test at its first appearance.

- In line 405, author should provide a percentage that is significantly higher.

We thank the reviewer for this observation. We agree that specifying the participation percentages improves the clarity and impact of the finding. In response, we have revised the sentence to report the actual participation rate associated with automatic FOBT delivery, based on the data from Vanaclocha-Espi et al. [38].

- Line 660, it should be “Contributions” instead of “Patents”

We have corrected this heading and replaced “Patents” with “Contributions”, which is the appropriate section title for this manuscript.

Round 2

Reviewer 1 Report

Comments and Suggestions for Authors

This manuscript has been sufficiently improved to warrant publication in Cancers.